# Cognition in older adults in Uganda: Correlates, trends over time and association with mortality in prospective population study

**Josephine E. Prynn**[1], **Calum Davey**[2], **Daniel Davis**[3], **Hannah Kuper**[2]*,
**Joseph Mugisha**[4], **Janet Seeley**[4,5]

**1** Institute of Cardiovascular Sciences, University College London, London, United Kingdom, **2** Faculty of Epidemiology and Population Health, London School of Hygiene and Tropical Medicine, London, United Kingdom, **3** MRC Unit for Lifelong Health and Ageing, University College London, London, United Kingdom, **4** MRC/UVRI & LSHTM Uganda Research Unit, Entebbe, Uganda, **5** Faculty of Public Health and Policy, London School of Hygiene and Tropical Medicine, London, United Kingdom

* hannah.kuper@lshtm.ac.uk

**Data Availability Statement:** The Wellbeing of Older People Study data are available through the WHO Multi-Country Studies Data Archive: https://

## Abstract

Dementia is an important and growing issue in sub-Saharan Africa, but epidemiological data are lacking. Risk factors may differ from other regions due to high stroke incidence and HIV prevalence. Understanding the epidemiology of cognition in older adults in Africa is crucial for informing public health strategies to improve the lives of people with dementia and their carers. The Wellbeing of Older People Study in Uganda is an open cohort of adults aged 50 + with and without HIV, established in 2009. Detailed socio-demographic and health data have been collected at four waves spanning 10 years, including cognitive assessment using internationally validated WHO-recommended tests: verbal recall, digit span, and verbal fluency. Mortality data was collected until the end of the fourth wave (2019). We examined associations of low baseline cognition scores and changes in cognition score over time using random effects modelling, care needs of people with lower cognition scores, and the relationship between cognition score and mortality. Data were collected on 811 participants. Older age, lower educational attainment, lower socio-economic position, and extremes of BMI were associated with lower cognition scores. Cognition scores declined faster at older ages, but rate of decline was not associated with cardiovascular disease or HIV at baseline. People with lower cognition scores required more assistance with Activities of Daily Living, but mortality rates were similar across the range of cognition scores. The crucial next step will be to investigate types and presentation of clinical dementia in this cohort, so we can better understand the clinical relevance of these findings to inform public health planning.

## Introduction

Dementia has a profound effect on individuals, families, and societies. It is the leading contributor to disability among older people in low- and middle-income countries [1], where the number of people with dementia is predicted to increase from 27 million to 89 million between

apps.who.int/healthinfo/systems/surveydata/index.php/catalog.

**Funding:** This study was financially supported by the National Institute on Aging in the form of a grant (R01-AG044917) for the The Wellbeing of Older People Study. This study was also financially supported by NIHR Global Research Professorship in the form of a grant (NIHR301621) received by HK. This study was also financially supported by the Foreign Commonwealth and Development Office in the form of a PENDA grant received by CD. This study was also financially supported by Wellcome Trust, UK in the form of a grant (WT107467) received by DD. This study was also financially supported by Medical Research Council, UK in the form of a grant (MC_UU_00019/2) received by DD. This study was also financially supported by the National Health Service, UK in the form of salary for JP. This study was also financially supported by Medical research Council/ Uganda Virus Research Institute & London School of Hygiene & Tropical Medicine Uganda Research Unit Uganda in the form of salary for JM. This study was also financially supported by London School of Hygiene & Tropical Medicine, UK in the form of salary for JS. The specific roles of these authors are articulated in the 'author contributions' section. The funders had no role in study design, data collection and analysis, decision to publish, or preparation of the manuscript.

**Competing interests:** The authors have declared that no competing interests exist.

2015 and 2050 [2]. Effective public health planning requires understanding of the incidence and drivers of dementia in Africa, and how dementia impacts function and mortality. Appropriate strategies may aim to improve quality of life for people with dementia, such as enhancing access to services, reducing stigma, supporting carers, and addressing risk factors to reduce incident dementia in people currently in early and midlife. Data from African countries on dementia are lacking and inconsistent, with prevalence estimates ranging from 2.3% to 20% [3]. Consequently, epidemiological research on dementia and cognition in Africa has been prioritised by the World Health Organization (WHO) and the African Dementia Consortium [3, 4].

There are a number of ways in which the epidemiology of dementia may differ in sub-Saharan Africa to other world regions. There is limited evidence that established risk factors from other world regions are related to dementia in sub-Saharan Africa, including with respect to poor socioeconomic status, literacy and education, or lifestyle factors such as anthropometry, smoking and alcohol [3]. The relative importance of the dementia subtypes in sub-Saharan Africa may also differ to that in more intensively studied global areas due to a different risk factor landscape [3], such as a high incidence of stroke [5], but lower prevalence of smoking, obesity, and sedentary lifestyles [6]. Additionally, a large cohort of people living with HIV are now approaching older age in sub-Saharan Africa, thanks to widely available antiretroviral therapy (ART) [7]. HIV may increase the rate of vascular ageing and neurodegenerative processes, and thus may be a significant contributor to dementia incidence [7, 8]. The association of poor cognition with disability and ability to perform activities of daily living is also not well established in Africa, yet critical for determining carer burden and need for support [3].

Most of the insights into risk factors for poor cognition in Africa are derived from cross-sectional studies [3]. Yet without capturing longitudinal change in cognition over time, interpreting associations is limited by reverse causality, mis-classification of people with long-term static low cognition scores, and survival bias through under-estimating dementia incidence due to differentially higher mortality. Dementia is a risk factor for death across many contexts, increasing mortality rates 5.9-fold [9]. This association may arise for a variety of reasons, including that people with dementia are on average older or living in poverty, on account of a higher prevalence of risk factors (e.g. malnutrition), or due to the health condition underlying the dementia (e.g. cardiovascular disease (CVD) or HIV). However, there are no data from Uganda, and little from other African research sites [10, 11].

The Wellbeing of Older People Study (WOPS) in Uganda provides the rare resource of 10 years' of cognition data obtained from adults aged 50 and over, with and without HIV. It contains detailed socio-demographic, clinical, disability, and mortality information. The aim of this study is to improve understanding of the epidemiology of cognition in Africa by describing the 1) correlates of baseline cognition scores (including sociodemographic, CVD, HIV, disability), 2) trends in cognition scores over time (including whether this varied by CVD or HIV status), and 3) association of cognition scores with mortality in a cohort of older Ugandan people.

## Methods

### Ethics statements

We obtained ethical approval from the Uganda Virus Research Institute, Uganda National Council for Science and Technology, and the London School of Hygiene and Tropical Medicine. All participants had given thumb-printed or written informed consent before data collection.

## Study setting and background

WOPS is run by the MRC/UVRI and LSHTM Uganda Research Unit, as part of the WHO Study on Global Ageing and Adult Health (SAGE). It is an open cohort of adults aged 50 established in 2009 in southwestern Uganda. At baseline, participants were selected randomly for enrolment from two pre-existing cohorts. The first cohort, in a rural site, recruited participants mainly within the General Population Cohort (GPC) located in Kalungu district in rural south-western Uganda. The GPC was established in 1989 to study the epidemiology of HIV, and there are annual demographic and serological suverys in the GPC. Some additional participants were recruited from The AIDS Support Organisation (TASO) in Makasa district. The second cohort is from Entebbe (Wakiso district, peri-urban) selected to include participants with and without HIV aged ≥50. The Entebbe cohort was established as an open cohort study that followed HIV-infected participants from 1994 to 2009. Additional participants were included from TASO in Entebbe.

WOPS consisted of HIV-positive and -negative people aged 50 years and older divided in five groups: 1) have an adult child who died of AIDS; 2) have an adult child who is living with HIV and on ART; 3) have no child with HIV/AIDS and are not infected with HIV themselves (comparison group); 4) are HIV infected and on ART for at least one year; and 5) are HIV infected and not on ART. Each group comprised about 100 respondents with half of the respondents recruited from the rural site and half from the urban site. At both sites, a list of all older people was first drawn based on the surveillance field site census records and nested surveys. For each study group about 100 people were randomly selected from these study lists. Refusals were less than 1%. Only residents were eligible for inclusion. The detailed methodology is described elsewhere [12, 13]. All WOPS cohort participants were eligible for inclusion in this study.

## Data collection

Data were collected in 4 waves: Wave 1 in 2009–2010, Wave 2 in 2012–2013, Wave 3 in 2015–2016, and Wave 4 in 2018–2019. Data were collected throughout the year until all study participants were covered. Participants were seen at home by interviewers trained in working with older people. Data collection comprised questionnaires on sociodemographic and health details, anthropomorphic measures including height, weight, and waist and hip circumference, blood pressure, and cognitive testing. Cognition was assessed at each Wave using WHO-recommended tests validated across different international sites [14]:

1. Verbal recall: a 10-word learning task where participants were read a list of 10 familiar words in Luganda (the local language) and asked to recall them three times, prompted at each attempt. Participants were asked to recall the list again after 10–15 minutes. At both points, the interviewer recorded the total number of verbs recalled correctly in one minute, number of words the participant substituted and the total number of words the participant failed.

2. Digit span forwards and backwards: recall of increasing numbers of digits forwards and then backwards

3. Verbal fluency: the number of animals a participant could name in one minute

If a participant was not found at home at follow-up, interviewers would return another time if other household members or neighbours confirmed the participant was still living at that address. If the participant had moved nearby, interviewers would proceed to their new address. If it was reported that they had died, date of death would be noted, and a verbal autopsy performed with consent of the next of kin.

## Variables

**Exposures.** Exposures of interest included sociodemographic details (age, urban or rural residence, educational attainment, marital status), behavioural risk factors (tobacco and alcohol use), CVD risk factors (body mass index, hypertension, and diabetes), CVD with end-organ damage (angina and stroke), and HIV.

Age was categorised into 50–69, 70–79, 80–89, ≥90. Socioeconomic position (SEP) was operationalised using principal component analysis of housing materials, household ownership of durable assets (e.g. bicycle, radio), and ownership of livestock, to create one overall score which was then categorised by quintiles. Educational attainment was dichotomised into self-reported "no formal education" and "any formal education", due to small numbers in older age groups with more than primary education. Participants were classified as having hypertension if they had a systolic blood pressure ≥140mmHg, a diastolic blood pressure ≥90mmHg, or reported taking antihypertensive medication. Body mass index (BMI) was categorised as $<18.5\,\mathrm{kg/m^2}$ (underweight), $18.5$–$24.9\,\mathrm{kg/m^2}$ (healthy weight), $25$–$29.9\,\mathrm{kg/m^2}$ (overweight), and $\geq30\,\mathrm{kg/m^2}$ (obese). Angina and stroke were defined as either an existing clinician diagnosis or a suggestive history at interview (chest pain on exertion for angina; episode of hemiparesis or hemisensory loss lasting at least 24 hours for stroke) and combined into a single variable of CVD. HIV status was confirmed at Wave 1 using rapid diagnostic tests and then all HIV-negative participants were retested at every subsequent round.

We measured disability using the WHODAS 2.0 12-item score [15], which asks about difficulties including standing, taking care of household responsibilities, and participating in community activities due to health conditions. For each of 12 questions, participants can answer on a scale of 1 to 5 from no difficulty to extreme difficulty/cannot do. The total score out of 60 was transformed into a score out of 100, for comparability with other studies.

**Cognition.** Participants had three separate scores for cognition (verbal recall, digit span, and verbal fluency). We assessed for correlation between the different components of cognition, and as correlation coefficients were low ($<0.5$) between components, they were treated as independent and combined into a single score. We calculated z-scores for each component and created an overall score of the mean of the 3 component scores. This method was chosen so that the very small number of participants missing 1 or 2 components could still be included. Cognition score was divided into quintiles to examine trends and associations with other variables. This same process was used to create cognition scores at the different Waves. The baseline cognition score was measured at Wave 1, Wave 2 or Wave 3, depending on timing of cohort entry of the participant.

## Statistical analyses

Cognition score was treated initially as a continuous variable and multivariable linear regression was used to assess the relationship between exposures (independent variables) and cognition score (dependent variable) at baseline including all participants from Wave 1, and new participants to Waves 2 and 3. All exposures associated with cognition score at univariate analysis were included in the multivariable model.

We used random-effects modelling to estimate the relationship of first CVD (defined as stroke or angina, indicating end-organ damage) and then HIV (independent variables), with change in cognition score with age (dependent variable). This model allows analysis of repeated measures of cognition score in the same individual at different ages, and accounts for clustering of data within individuals. The exposures (CVD then HIV) were used as interaction terms with age group, to assess whether cognitive score changes with age (dependent variable) were affected by each exposure (independent variable), adjusted for sex and educational

attainment (covariates). The mean cognition score predicted by the model was then plotted in a line graph by age group and exposure.

The WHODAS 2.0 12-item disability scale was introduced in Wave 2. For Wave 2 participants, we calculated the mean disability score by baseline cognition group, and the percentage of people in each cognition score group requiring assistance with instrumental Activities of Daily Living (iADLs) including buying food, cooking, fetching water, and agricultural activities, and with Activities of Daily Living (ADLs) including bathing, eating, dressing, and getting out of bed.

Finally, we used multivariable Poisson regression analysis to estimate the association of mortality rates (dependent variable) and baseline cognition score (independent variable). As this was an open cohort, individuals contributed exposure time from the date of their first interview until the date of their last interview or death, whichever came first. Participants lost to follow-up were censored at a date halfway between their last interview date and the median date of data collection for the following wave. Model 1 was adjusted for age only; Model 2 was adjusted for all variables found to confound the relationship between cognition and mortality (covariates).

The small amount of missing data allowed us to use a complete case analysis.

## Results

Data were collected from 509 participants at Wave 1, with 126 additions at Wave 2 and 176 at Wave 3, giving a total of 811 participants (*Fig 1*). Overall, 122 participants died and 199 were lost to follow-up.

Of the 811 participants with baseline cognition data obtained between Waves 1 to 3, 47% lived in a rural area and 60% were women; 42% were aged 50–59, 9% were aged 80 or over, and 79% had had some formal education (*Table 1*, *S1 Table*). Prevalence of hypertension was 55% and diabetes was 5%; 5% of people had had a stroke, 23% experienced angina, and 26% had CVD (either stroke or angina). Sixty percent were HIV positive. In a multivariable linear regression analysis, increasing age, lower educational attainment, lower socio-economic position, and having a BMI below 18.5 kg/m$^2$ or above 30 kg/m$^2$ were associated with a lower baseline cognition score. Neither CVD risk factors, overt CVD, nor HIV status were associated with cognition at baseline.

Average cognitive score decreased over time, with more rapid decline after age 80 (*Fig 2*). For an average man, the expected 5-year cognitive score decline would be 0.04 SD from age 50 and 0.23 SD from age 70. We saw no difference in cognitive score decline with age in people with CVD (p = 0.46) or HIV (p = 0.48) compared with those without those conditions. Similarly, cognitive score decline with age was not associated with educational attainment or SEP (*S1 and S2 Figs*).

Mean disability score increased with decreasing cognition score at baseline, and for those in the bottom quintile of cognition, 37.2% needed assistance with iADLs (test for trend, p = 0.006), and 22.3% needed help with ADLs, compared to 17.9% and 9.5% respectively among adults in the top quintile (test for trend, p = 0.04) (*Table 2*).

798 participants were followed up over 4889 person-years, and 123 people died. Over the 10-year follow up, 26% of participants from Wave 1 were lost to follow-up (*S3 Table*). The observed gradient in unadjusted mortality rate with decreasing cognition score at baseline was largely explained after adjustment for sex, socio-economic position and educational attainment (p = 0.06) (*Table 3*).

## Discussion

We found that among this Ugandan population, increasing age, lower educational attainment, lower SEP, and extremes of BMI were associated with a lower baseline cognition score.

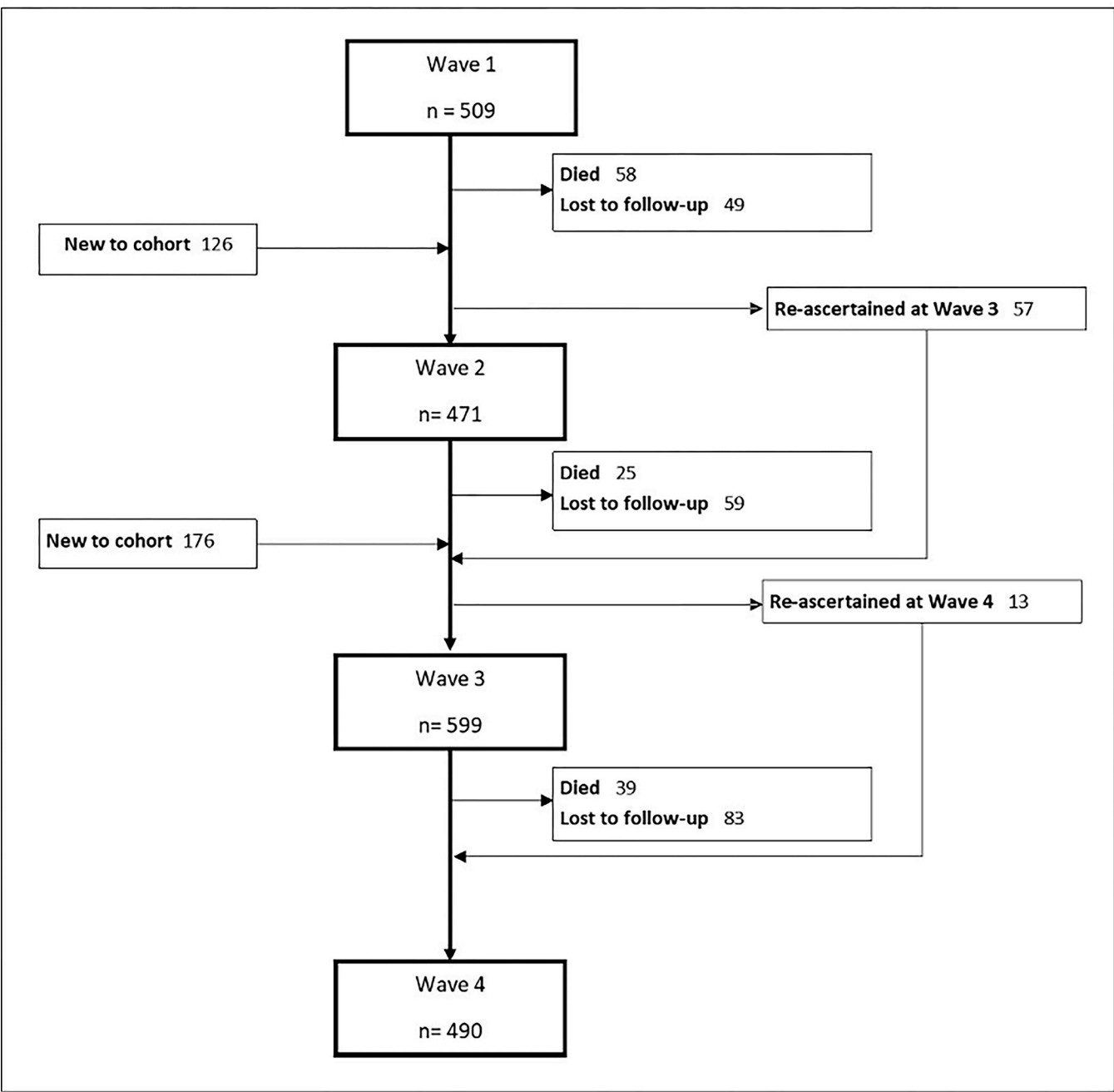

**Fig 1. Flow chart of study participants.**

Participants with a lower cognition score had higher care needs. Over time, the cognition score declined with age, with more rapid changes after age 80. Neither CVD nor HIV was associated with increased decline in cognition scores with age. We did not find mortality rates to be higher among people with lower cognition scores.

Or findings are in keeping with existing global health literature [16]. The relationship between cognition and education is often attributed to the 'cognitive reserve' theory, where the clinical features of cognitive decline are delayed in people with more education, despite

**Table 1. Baseline characteristics of participants and correlates of baseline cognition score.**

| | | N (%) | Mean baseline cognition score[1] | Linear regression coefficient[2] (95% CI) | P-value |
|---|---|---|---|---|---|
| **Overall** | | 811 | 0.01 (range: -2.78; 2.99) | | |
| **Wave at recruitment** | 1 | 509 (63) | -0.007 | Ref | 0.05 |
| | 2 | 126 (16) | 0.10 | -0.16 (-0.31;-0.01) | |
| | 3 | 176 (22) | -0.004 | -0.11 (-0.23;0.02) | |
| **Residence** | Rural | 377 (47) | -0.09 | Ref | 0.13 |
| | Urban | 434 (54) | 0.10 | 0.08 (-0.03;0.19) | |
| **Sex** | Male | 323 (40) | 0.09 | Ref | 0.29 |
| | Female | 488 (60) | -0.04 | -0.07 (-0.19;0.06) | |
| **Age group** | 50–59 | 339 (42) | 0.28 | Ref | <0.0001 |
| | 60–69 | 242 (30) | 0.004 | -0.24 (-0.36;-0.12) | |
| | 70–79 | 160 (20) | -0.25 | -0.47 (-0.61;-0.33) | |
| | 80+ | 70 (9) | -0.67 | -0.85 (-1.03;-0.66) | |
| **Education[3]** | No formal education | 168 (21) | -0.42 | Ref | <0.0001 |
| | Any formal education | 642 (79) | 0.12 | 0.32 (0.19;0.45) | |
| **Marital status[3,4]** | Married/cohabiting | 285 (35) | 0.17 | Ref | 0.44 |
| | Not married | 525 (65) | -0.07 | -0.05 (-0.17;0.07) | |
| **Socio-economic position** | 1 (Lowest) | 173 (22) | -0.31 | Ref | <0.0001 |
| | 2 | 173 (22) | -0.03 | 0.26 (0.11;0.41) | |
| | 3 | 173 (22) | 0.06 | 0.23 (0.08;0.39) | |
| | 4 | 170 (21) | 0.20 | 0.35 (0.19;0.51) | |
| | 5 (Highest) | 110 (14) | 0.23 | 0.4 (0.22;0.58) | |
| **Tobacco use [3]** | Never used tobacco | 128 (16) | -0.15 | Ref | 0.88 |
| | Current tobacco use | 130 (16) | 0.05 | 0.01 (-0.16;0.19) | |
| | Previous tobacco use | 552 (68) | 0.04 | -0.02 (-0.17;0.12) | |
| **Ever consumed alcohol[3]** | No | 593 (73) | 0.009 | Ref | 0.19 |
| | Yes | 217 (27) | 0.02 | 0.08 (-0.04;0.19) | |
| **BMI (kg/m[2])** | <18.5 | 115 (14) | -0.23 | -0.17 (-0.32;-0.02) | 0.02 |
| | 18.5- | 486 (60) | 0.04 | Ref | |
| | 25- | 127 (16) | 0.18 | 0.07 (-0.07;0.21) | |
| | 30+ | 83 (10) | -0.11 | -0.14 (-0.31;0.03) | |
| **Hypertension** | No | 445 (55) | 0.26 | Ref | 0.51 |
| | Yes | 366 (45) | -0.009 | 0.03 (-0.07;0.14) | |
| **Diabetes[3]** | No | 771 (95) | 0.005 | Ref | 0.83 |
| | Yes | 38 (5) | 0.16 | -0.03 (-0.27;0.21) | |
| **Stroke[6]** | No | 771 (95) | 0.007 | Ref | 0.33 |
| | Yes | 40 (5) | 0.07 | 0.11 (-0.12;0.34) | |
| **Angina[3,6]** | No | 624 (77) | 0.02 | Ref | 0.52 |
| | Yes | 186 (23) | 0.004 | 0.04 (-0.08;0.16) | |
| **CVD** | No | 598 (74) | 0.02 | Ref | 0.52 |
| | Yes | 212 (26) | -0.006 | 0.04 | |
| **HIV status[7]** | HIV negative | 323 (40) | -0.10 | Ref | 0.49 |
| | HIV positive | 483 (60) | 0.09 | -0.05 (-0.18;0.09) | |

1. Higher scores represent better cognition

2. Linear regression controlled for sex, age group, residence, marital status, education, socioeconomic position, tobacco use, and BMI

3. Missing data for 1 participant

4. Not married includes: never married, separated, divorced, and widowed

5. Socio-economic position calculated using primary component analysis; missing data for 2 participants

6. Stroke and angina defined as clinician diagnosis or suggestive history

7. Missing data for 4 participants

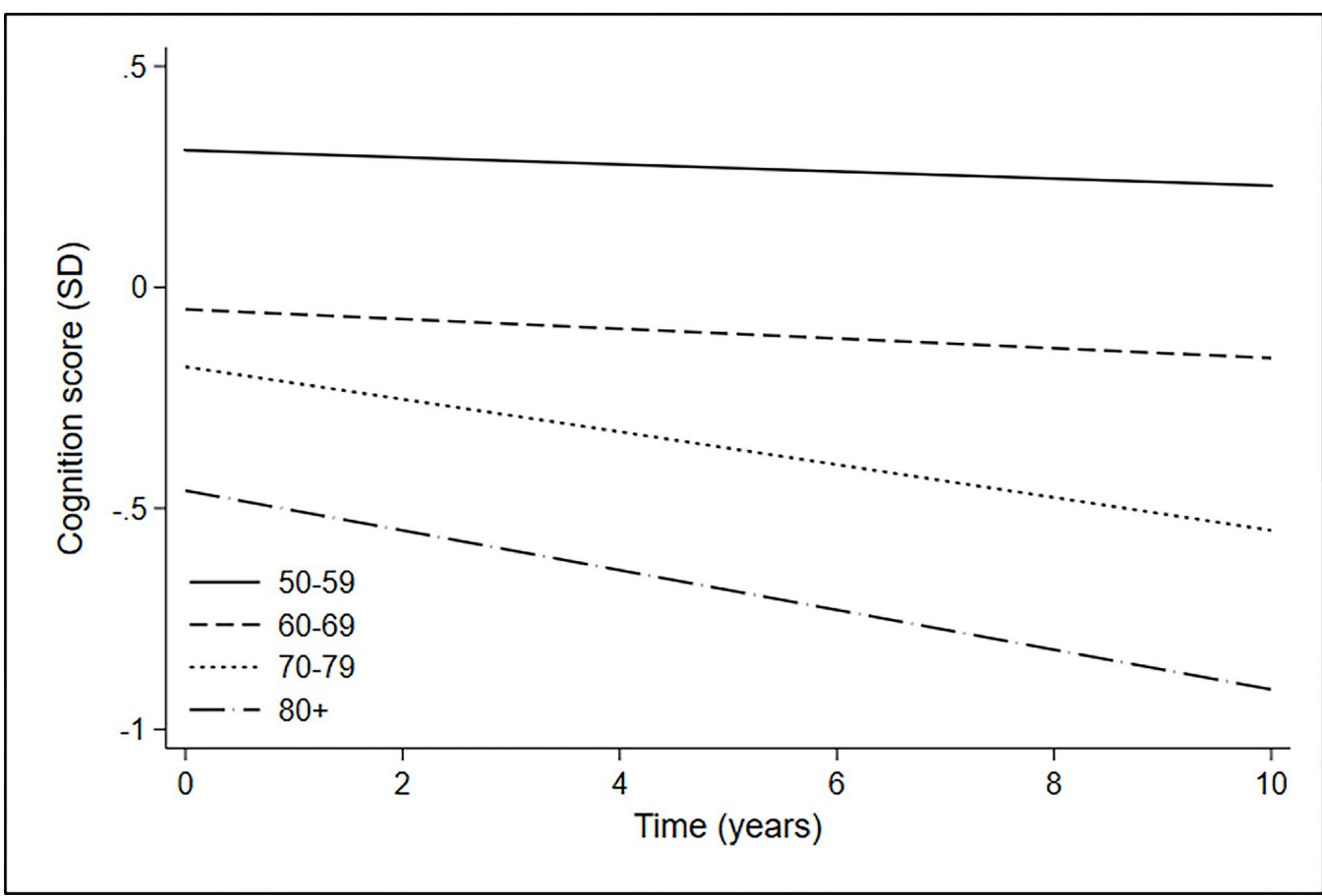

**Fig 2. Cognition scores over time by age group, adjusted for sex, SEP, educational attainment and BMI.**

underlying disease pathology [16]. The relevance of this has been questioned in some sub-Saharan Africa settings, where formal education was historically less developed, and the protective effect of education has not been consistently observed [2]. Furthermore, in low-literacy

**Table 2. Disability score and care needs by baseline cognition score.**

| | | n | Mean disability score[1] (Range 0–100) | Care needs | | | | | |
|---|---|---|---|---|---|---|---|---|---|
| | | | | Instrumental Activities of Daily Living (iADLs) [2] | | | Activities of Daily Living (ADLs) [3] | | |
| | | | | n | % (95% CI) | p-value[4] | n | % (95% CI) | p-value[4] |
| Cognition score groups | 5 (highest) | 95 | 21.4 | 17 | 17.9 (11.4;27.0) | 0.006 | 9 | 9.5 (5.0;17.3) | 0.04 |
| | 4 | 93 | 22.9 | 16 | 17.0 (10.7;26.0) | | 8 | 8.5 (4.3;16.1) | |
| | 3 | 96 | 24.0 | 22 | 22.9 (15.6;32.4) | | 14 | 14.6 (8.8;23.2) | |
| | 2 | 91 | 29.0 | 27 | 29.3 (20.9;39.5) | | 16 | 17.4 (10.9;26.6) | |
| | 1 (lowest) | 94 | 34.6 | 35 | 37.2 (28.1;47.4) | | 21 | 22.3 (15.0;31.9) | |

1. Disability score calculated using WHODAS 2.0 12-item scale; a higher score represents greater disability

2. iADLs include buying food, cooking, fetching water, agricultural work

3. ADLs include bathing, eating, dressing, getting out of bed, using the toilet, taking medicines

4. Chi-square test for trend

**Table 3. Poisson regression analysis of mortality by cognition score at baseline.**

| | | N of deaths | Person years (PY) | Mortality rate (per 1000 PY) | Model 1: adjusted for age group | | Model 2: fully adjusted[1] | |
|---|---|---|---|---|---|---|---|---|
| | | | | | RR | p-value | RR | p-value |
| Cognition score groups | 5 (highest) | 13 | 1041 | 12.5 | Ref | 0.02 | Ref | 0.06 |
| | 4 | 12 | 1016 | 11.8 | 0.69 (0.31;1.51) | | 0.66 (0.30;1.47) | |
| | 3 | 28 | 918 | 30.5 | 1.77 (0.91;3.44) | | 1.71 (0.87;3.36) | |
| | 2 | 29 | 989 | 29.3 | 1.53 (0.79;2.98) | | 1.28 (065;2.51) | |
| | 1 (lowest) | 41 | 925 | 44.3 | 1.58 (0.82;3.08) | | 1.40 (0.71;2.78) | |

1. Adjusted for age group, sex, socio-economic position, and educational attainment

contexts, word-learning tests may under-estimate cognitive ability in people without formal education, which has implications for classificiation of low cognition or dementia [17].

The lack of association between cognition scores and CVD or risk factors shown in this study is similar to results from studies of older adults in South Africa, Tanzania and Nigeria [18–20], and in contrast to findings from high-income countries [21]. This may reflect survival bias, whereby people with CVD die of other complications before cognitive decline, or are dying rapidly after their cognition starts declining. However, using clinician-diagnosed dementia as an outcome, an association between both hypertension and peripheral arterial disease was noted in Central African Republic and the Republic of the Congo [22, 23].

HIV has long been associated with poor cognitive outcomes [8], and has been postulated to accelerate Alzheimer's disease and vascular disease processes [7]. However, it is not clear from existing literature what effect, if any, chronic and well-controlled HIV has on cognition with ageing, due to difficulties disentangling the effects of comorbidity, ART, the impact of SEP and education on performance in cognition testing [24, 25], and a paucity of studies in older people [7]. We found no convincing association between HIV and cognition score at baseline, nor with cognitive decline. However, the numbers of people with HIV in the oldest age groups were very small, limiting the interpretability of these results. The coming decades will allow closer scrutiny of the impact of chronic HIV infection on the ageing brain, as younger HIV positive cohorts age into the 80+ age groups.

Poor cognition is known to be associated with high care needs globally [1], and we found that nearly 40% of those in the lowest cognition score group needed assistance with iADLs and 22% needed help with ADLs. These results are comparable to other studies in the continent: in a study of middle-aged and older adults in South Africa, around 20–40% of people in the lowest group of cognition score had an ADL impairment [26], and in Tanzania, 41% of people with dementia had moderate or severe disability [27]. Strain among carers of people with dementia has been well-documented in the SSA setting, including in Uganda [28], and carer support packages relevant to African settings may help to address this issue [29].

The relationship between low cognition score and mortality rates was largely explained by SEP and education. These findings contrast with worldwide evidence that dementia is a risk factor for death, including from sub-Saharan Africa [10, 11, 30, 31]. In Tanzania, older adults reporting subjective memory problems had an odds ratio of mortality of 3.0 compared to those who did not, adjusted for age, sex, disability, BMI, and blood pressure [11]. In Nigeria, people with diagnosed dementia had a mortality rate ratio 1.5 times that of people without dementia, adjusted for age, sex, socio-economic position, occupation, and pre-dementia cognitive function [10]. Participants in our study were categorised by cognition score rather than diagnosed dementia, but we could not demonstrate that any relationship between cognition

and mortality extends throughout the cognitive spectrum–a key finding of cognitive epidemiology studies from high-income settings [30, 31]. The mechanism by which higher intelligence or cognition might be associated with a survival advantage is not fully understood [30]. Plausibly the skills most beneficial for longevity in a low-income setting such as Uganda may differ from in an industrialised, high-income setting, and could be less well captured by cognitive screening tools.

## Strengths and limitations

This study makes use of longitudinal data, with multiple waves of data collection using identical cognitive screening tools at each wave. The study population was enriched with people with HIV, increasing the power of analyses to describe differences in cognition score between people with and without HIV. The population sampling frame had the advantage of obtaining clinical symptoms for angina and stroke, not requiring participants to have had previous contact with health services.

This study had a number of limitations. The key concern is that our outcome measure was cognition score based on screening tools, rather than clinical dementia diagnosis. Screening tools are valuable as they are quick to administer, do not require clinical training, and are therefore scalable for longitudinal population studies. Limitations of the cognitive test used include that while it would effectively measure memory, concentration and language function, there was no test of visuo-spatial ability or executive functioning. Furthermore, we could not distinguish participants with dementia from those with (lifelong) low-norm non-pathological cognition scores, limiting our conclusions on dementia risk factors and progression. We also combined three cognition tests into one score using an approach consistent with other studies [32]. However, a standardised approach does not exist for how this combination should be done, which raises concerns about the validity and interpretability of the new measure. Additionally, performance in cognitive tests can be affected by factors other than true cognitive capability, including education and SEP [24], which limits our ability to determine differences. While the WHO recommended tool for the SAGE studies is a culturally-sensitive tool which was pilot-tested in India, Tanzania and Ghana [14], this was its first use in Uganda, and validation studies with a clinician diagnosis of dementia are awaited. Another issue is that we combined participants from different cohorts and loss to follow-up was significant, with 26% of Wave 1 participants lost by Wave 4, though this group had a similar distribution of age, sex, and baseline cognition score. People without cognitive decline may have been more likely to move away for new opportunities such as work, or conversely, people with cognitive decline may have been missed in the survey if they were unwell and seeking healthcare, or if they were hidden by community members due to stigma. Finally, while many of the health-related exposures were objectively tested (e.g. HIV, BMI, blood pressure) ascertainment of elements of the medical history were by self-report of symptoms (e.g. angina, stroke). Participants with poor cognition may have difficulty recalling this information, leading to measurement error in the exposures, which would bias the associations between risk factors and the outcome towards the null.

## Conclusion

We generated a new amalgamated cognition score to describe the epidemiology of cognition among older people in a cohort in Uganda. People with low cognition scores were more likely to be older, less well-educated, and with a lower socio-economic position. Disability and care needs increased with decreasing cognition score, though lower cognition scores were not associated with mortality. We found no association between CVD or HIV and baseline cognition

score or changes in cognition score. The crucial next step will be to investigate types and presentation of clinical dementia in this cohort, so we can better understand the clinical relevance of these findings to inform public health planning.

## Supporting information

**S1 Table. Detailed baseline characteristics by wave and sex.**
(DOCX)

**S2 Table. Mean score and correlates of each cognition component.**
(DOCX)

**S3 Table. Baseline characteristics stratified by whether participants completed the follow up period.**
(DOCX)

**S1 File. Questionnaire on inclusivity.**
(DOCX)

**S1 Fig. Cognition scores over time by educational attainment, adjusted for age, sex, socio-economic position, and BMI.**
(TIFF)

**S2 Fig. Cognition scores over time by socio-economic position (SEP), adjusted for age, sex, educational attainment, and BMI.**
(TIFF)

## Author Contributions

**Conceptualization:** Joseph Mugisha, Janet Seeley.

**Data curation:** Joseph Mugisha.

**Formal analysis:** Josephine E. Prynn.

**Funding acquisition:** Joseph Mugisha, Janet Seeley.

**Project administration:** Joseph Mugisha.

**Resources:** Joseph Mugisha, Janet Seeley.

**Writing – original draft:** Josephine E. Prynn.

**Writing – review & editing:** Calum Davey, Daniel Davis, Hannah Kuper, Joseph Mugisha, Janet Seeley.

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
