## [Decision Letter · Decision Letter 0]

7 Feb 2023

PGPH-D-23-00011

Longitudinal change in cognition in older adults in Uganda: a prospective population study

Dear Dr. Kuper,

Thank you for submitting your manuscript to PLOS Global Public Health. After careful consideration, we feel that it has merit but does not fully meet PLOS Global Public Health’s publication criteria as it currently stands. Therefore, we invite you to submit a revised version of the manuscript that addresses the points raised during the review process.

We look forward to receiving your revised manuscript.

Kind regards,

Paolo Angelo Cortesi, PhD

Academic Editor

Journal Requirements:

3. Please provide separate figure files in .tif or .eps format.

4. In the online submission form, you indicated that your data will be submitted to a repository upon acceptance.  We strongly recommend all authors deposit their data before acceptance, as the process can be lengthy and hold up publication timelines. Please note that, though access restrictions are acceptable now, your entire data will need to be made freely accessible if your manuscript is accepted for publication. This policy applies to all data except where public deposition would breach compliance with the protocol approved by your research ethics board. If you are unable to adhere to our open data policy, please kindly revise your statement to explain your reasoning and we will seek the editor's input on an exemption. Please be assured that, once you have provided your new statement, the assessment of your exemption will not hold up the peer review process. 

Additional Editor Comments (if provided):

Reviewers' comments:

Reviewer's Responses to Questions

**Comments to the Author**

1. Does this manuscript meet PLOS Global Public Health’s publication criteria? Is the manuscript technically sound, and do the data support the conclusions? The manuscript must describe methodologically and ethically rigorous research with conclusions that are appropriately drawn based on the data presented.

Reviewer #1: Yes

Reviewer #2: Partly

2. Has the statistical analysis been performed appropriately and rigorously?

Reviewer #1: I don't know

Reviewer #2: No

3. Have the authors made all data underlying the findings in their manuscript fully available (please refer to the Data Availability Statement at the start of the manuscript PDF file)?

Reviewer #1: Yes

Reviewer #2: No

4. Is the manuscript presented in an intelligible fashion and written in standard English?

Reviewer #1: Yes

Reviewer #2: Yes

5. Review Comments to the Author

Reviewer #1: The study is interesting and relevant. It is not clear if the Wellbeing of Older People Study (WOPS) was conducted throughout the year or during specific times of the year. That way, they can determine weather conditions as influencing results as well.

The next steps can also be to explore other factors such as the potential links between precipitation and age-related cognitive decline. And the role of diet in cognitive decline among such populations.

Reviewer #2: Longitudinal change in cognition in older adults in Uganda: a prospective population study

Title.

The paper is about the association of the changes in cognition with HIV or mortality or CVD -the independent variable is not specified-. Therefore, the title should change after reviewing all comments.

Introduction.

It is necessary to add quantitative data to support the statements, for example, the prevalence and incidence of dementia in Africa or the known risk factors (like HIV) for dementia, including their odds ratio values.

The analysis contemplates certain variables not mentioned in the background, so there is any theoretical support for their inclusion in the article (disability, socioeconomic status, education).

The purpose of the article needs to be clarified. First, the dependent and independent variables must be identified. Also, the objective is poorly structured because it has two outcomes (dead and cognitive deterioration -no dementia-) and three dependent variables (CVD, HIV, disability).

Material and Methods.

No inclusion, exclusion, or elimination criteria are mentioned.

The WHO cognition test needs to be sufficiently explained; the scores and assessment must be understood. Moreover, this test cannot diagnose dementia (just cognitive function), so it is impossible to analyze dementia risk factors.

It has yet to explain how authors manage the changes down the cohort; for example, there is no information if the measurement of the variables from the first or the last wave or if the cognitive assessment was just for the ones from the first wave. Moreover, figures 2 and 3 (the title of figure 3 is wrong) do not specify if the values are from the follow-up observations from the first wave.

It is necessary to differentiate between the dependent and independent variables and the covariates. Also, a more accurate explanation of the socioeconomic position and educational attainment variable construction is required.

The way the cognitive variable was treated is confusing (described on page 5); it seems arbitrary. Also, including participants that do not have all the components from the cognitive questionnaire and transforming the original disability score of 60 to 100 is methodologically incorrect.

A better redaction of the statistical analysis section is required, step by step.

The linear regression technique can only be used if the independent variable is explicit. Moreover, mortality cannot be estimated with Poisson analysis (page 6).

Results.

This section should be modified, taking into consideration the previous comments.

It needs to be clarified the way information is organized. Please, do not combine descriptive and association results (table 1, 2, 3).

On page 9, it needs to be clarified why the authors calculated the person-years value. Additionally, talking about the crude mortality rate needs to be corrected (crude mortality is the mortality of the total population, not a sample).

Discussion.

This section should be modified, taking into consideration the previous comments.

The discussion is focused on the relationship between the findings and dementia. However, cognitive function decline could have a lot of reasons (like aging and traumas), so I see an author bias.

In the limitations section, the authors said that one of the article's strengths is that with this data, they can analyze the differences between populations with and without HIV, which is not valid. Additionally, one of the limitations listed is that with the scale used, it is not possible to assess dementia or related risk factors, so -by knowing this from the beginning- there is a contradiction with the objective of the article.

Conclusion.

This section should be reviewed, taking into consideration all the comments above.

General comments.

Review English redaction.

Review the correct definitions of acronyms.

Figure 1 does not have a title.

Figure 3 has the wrong title.

6. PLOS authors have the option to publish the peer review history of their article (what does this mean?). If published, this will include your full peer review and any attached files.

**Do you want your identity to be public for this peer review?** For information about this choice, including consent withdrawal, please see our Privacy Policy.

Reviewer #1: **Yes: **Funmi Akindejoye

Reviewer #2: **Yes: **CArmen García-Peña

---

## [Decision Letter · Decision Letter 1]

17 May 2023

PGPH-D-23-00011R1

Cognition in older adults in Uganda: correlates, trends over time and association with mortality in prospective population study

Dear Dr. Kuper,

Thank you for submitting your manuscript to PLOS Global Public Health. After careful consideration, we feel that it has merit but does not fully meet PLOS Global Public Health’s publication criteria as it currently stands. Therefore, we invite you to submit a revised version of the manuscript that addresses the points raised during the review process.

We look forward to receiving your revised manuscript.

Kind regards,

Paolo Angelo Cortesi, PhD

Academic Editor

Journal Requirements:

Additional Editor Comments (if provided):

Reviewers' comments:

Reviewer's Responses to Questions

**Comments to the Author**

1. If the authors have adequately addressed your comments raised in a previous round of review and you feel that this manuscript is now acceptable for publication, you may indicate that here to bypass the “Comments to the Author” section, enter your conflict of interest statement in the “Confidential to Editor” section, and submit your "Accept" recommendation.

Reviewer #2: (No Response)

2. Does this manuscript meet PLOS Global Public Health’s publication criteria? Is the manuscript technically sound, and do the data support the conclusions? The manuscript must describe methodologically and ethically rigorous research with conclusions that are appropriately drawn based on the data presented.

Reviewer #2: No

3. Has the statistical analysis been performed appropriately and rigorously?

Reviewer #2: No

4. Have the authors made all data underlying the findings in their manuscript fully available (please refer to the Data Availability Statement at the start of the manuscript PDF file)?

Reviewer #2: No

5. Is the manuscript presented in an intelligible fashion and written in standard English?

Reviewer #2: No

6. Review Comments to the Author

Reviewer #2: association with mortality in a prospective population study

Introduction

There is any theoretical support for including certain variables in the article (like disability, socioeconomic status, education, IALS, and ADLs). Also, a more extensive literature review is required to understand the problem addressed.

The aim still needs to be clarified. For example, the “cognitive score” seems to be the dependent variable, so what does “baseline cognition (including sociodemographic, CVD, HIV, disability)” means in the objective?

Again, it needs to be clarified the objective of the article. In the limitation section, the authors said that one of the strengths is the possibility of comparing persons with and without HIV. However, the authors should have made this comparison in the analysis and methodology. It seems a contradiction.

Methodology

No inclusion, exclusion, or elimination criteria are mentioned.

A more specific description of the cohort is required. Lines 97-106 of page 26 need to give more information to understand how the cohort was constructed. Moreover, the aggregate classification (numbers 1 to 5) needs to be clarified.

If cognition was measured at each wave, extensive detail is required of how the “cognition score” was constructed considering the time measures of cognition in every wave (specifying what happened with the observations that do not have the 4 cohort measures).

The outcome variable cannot be measured like that (cognition score). The arbitrary “combination” of the three different components of cognition is wrong (even if the authors specify it in the limitations). Indeed, it is not clear what this “cognition score” means. Again, including participants that do not have all the components from the cognitive questionnaire and transforming the original disability score of 60 to 100 is methodologically incorrect. So, authors cannot define low, middle, or high cognition. Consequently, results and conclusions do not have the support to be interpreted.

Analysis

Poisson regression analysis cannot calculate mortality rates.

The authors should have mentioned how they managed the loss of 26% of the sample.

Results

Separating descriptive and association results is better because it makes it simple to understand the findings. Therefore, leaving the descriptive and association results could be better.

Again, why the authors calculated and included the person-years value must be clarified. The reason given by the authors needs to explain it.

Discussion

Results cannot be interpreted due to the lack of support for the “cognition score” because the variable does not measure cognition. So, all the statements about the findings are incorrect (lines 221-222-223; 232; 234-235; 246-247; 259-260).

The strength list at the beginning of the limitations paragraph (page 33) must be corrected. The cohort needed more robust methodology support because it was the product of the fusion of diverse cohorts with different inclusion and exclusion criteria. Additionally, the authors said that it had a loss of follow-up of 26% of the sample.

This paper needs to provide correct and acceptable conclusions due to the methodology mistakes in the outcome variable and the inappropriate use of statistical analysis techniques.

7. PLOS authors have the option to publish the peer review history of their article (what does this mean?). If published, this will include your full peer review and any attached files.

**Do you want your identity to be public for this peer review?** For information about this choice, including consent withdrawal, please see our Privacy Policy.

Reviewer #2: **Yes: **Carmen Garcia-Peña

---

## [Editor Report · Decision Letter 2]

15 Sep 2023

Cognition in older adults in Uganda: correlates, trends over time and association with mortality in prospective population study

PGPH-D-23-00011R2

Dear Dr. Kuper,

We are pleased to inform you that your manuscript 'Cognition in older adults in Uganda: correlates, trends over time and association with mortality in prospective population study' has been provisionally accepted for publication in PLOS Global Public Health.

Best regards,

Darshini Govindasamy

Academic Editor